# Effects of Protective Surface Coating on Fluoride Release and Recharge of Recent Uncoated High-Viscosity Glass Ionomer Cement

**DOI:** 10.3390/dj10120233

**Published:** 2022-12-09

**Authors:** Nantawan Krajangta, Chayanee Dulsamphan, Tongjai Chotitanmapong

**Affiliations:** 1Division of Restorative Dentistry, Faculty of Dentistry, Thammasat University, Bangkok 10200, Thailand; 2Police General Hospital, Bangkok 10330, Thailand; 3Division of Pediatric Dentistry, Faculty of Dentistry, Thammasat University, Bangkok 10200, Thailand

**Keywords:** high-viscosity glass ionomer cement, surface coating, fluoride release, fluoride recharge

## Abstract

The ability to release and recharge fluoride is a property of glass ionomer cement materials, which is an advantage for patients with a high caries risk. The purpose of this study was to evaluate the amount of released and recharged fluoride in recent uncoated high-viscosity glass ionomer cement (Ketac^TM^ Universal Aplicap^TM^) with different surface coatings and at different time points. In total, 135 cylindrical-shaped specimens were equally divided into the following three groups: Ketac^TM^ Universal Aplicap^TM^, Ketac^TM^ Molar Aplicap^TM^, and Ketac^TM^ Fil Plus Aplicap^TM^. The different coatings performed on each group were as follows: uncoated, coated with Ketac^TM^ Glaze, and coated with G-Coat Plus^TM^. The amounts of released and recharged fluoride were measured at 24 h and at weeks 1, 2, 3, and 4. The recharging agent was a 1.23% APF gel. Ketac^TM^ Universal Aplicap^TM^ showed the highest released fluoride at all time points and the highest recharged fluoride at weeks 1, 2, and 3. Both the Ketac^TM^ Glaze- and G-Coat Plus^TM^-coated specimens presented significantly lower released and recharged fluoride ions than the uncoated group at all time points (*p* < 0.001). Coating with G-Coat Plus^TM^ significantly decreased the released and recharged fluoride compared to the coating with Ketac^TM^ Glaze at almost all time points (*p* < 0.05), except for weeks 1 and 2. The application of coating agents reduced the amount of released and recharged fluoride by the Ketac^TM^ Universal Aplicap^TM^.

## 1. Introduction

Glass ionomer cements (GICs) were first introduced in 1972 by Wilson and Kent [1]. The properties of GICs include a chemical bond to the tooth structure [2,3] and the release and recharge of fluoride. In addition, GICs present good biocompatibility [2], excellent coefficients of linear thermal expansion/contraction [2,4,5], and an excellent modulus of elasticity. However, a major disadvantage of GICs is their weak mechanical properties, such as low wear resistance, low fracture toughness [5,6,7], and sensitivity to moisture during the early stages of setting [4,7,8,9,10,11,12]. To overcome the limitations of conventional GICs, various modifications have been developed. By increasing the powder/liquid ratio and incorporating more reactive silicate particles [9,13,14], high-viscosity glass ionomer cements (HVGICs) were developed in 1990 [15].

HVGICs have improved mechanical properties [9,16] and faster setting reactions compared to GICs [6,13,14,16]. However, HVGICs are still sensitive to water absorption and dehydration during early setting reactions [4,8,11,17]. For this reason, the application of a protective surface coating is recommended [4,10,18,19]. Previous studies have reported that applying a surface coating could improve their mechanical properties, including their surface hardness [20,21,22], but decrease their fluoride release and recharge abilities [23,24,25,26,27,28].

The ability to release and recharge fluoride is one of the most interesting properties of glass ionomer cements (GICs), which results in a cariostatic effect and remineralization that benefits high-caries-risk patients [3]. Fluoride ions help to slow down the demineralization process and enhance the remineralization of enamel caries. The ability to recharge fluoride from an exogenous source is the key to maintaining fluoride levels as a fluoride reservoir, which can be recharged by topical fluoride, fluoride toothpaste, and mouthwash. In vitro exposure to topical fluoride can create a fluoride recharge potential [3].

In addition, GICs present chemical bonding to the tooth structure, good biocompatibility, an excellent modulus of elasticity, and excellent coefficients of linear thermal expansion and contraction [2,29]. However, the long setting time [16], weak mechanical properties, and sensitivity to water sorption or dehydration in their initial setting reaction [9] are the main disadvantages of GICs. By increasing the powder/liquid ratio and incorporating more reactive silicate particles [14], high-viscosity GICs (HVGICs) were developed for better mechanical properties [9]. Meanwhile, moisture sensitivity is still a problem for HVGICs [11]. The application of a protective surface coating acts as a barrier to water exchange and is recommended [3]. Some studies reported that applying the surface coating to some HVGICs improved their mechanical properties, such as their surface hardness [20], but decreased their fluoride release [23,24,25] and recharge capacities [24]. However, the application of a surface coating is mandatory for most commercially available HVGICs.

Ketac^TM^ Universal Aplicap^TM^ (3M ESPE, 3M ESPE, Deutschland GmbH, Neuss, Germany), an uncoated HVGIC, was first introduced in 2016, and the manufacturing company recommends that this material does not require coating steps. The material was claimed to be highly compressive with a high flexural strength because of a special filler composition and an accelerated setting reaction, resulting in reduced chair time and water sensitivity [14]. A recent study that evaluated the effects of the coating on the compressive strength, flexural strength, hardness, and color changes of this material found that applying the surface coating could improve the mechanical properties, such as the hardness. [30,31,32]. Nevertheless, limited publications have evaluated the effects of surface coatings on the releasing and recharging capacities of fluoride in this recent material. The purpose of this study was to evaluate the amount of released and recharged fluoride by the Ketac^TM^ Universal Aplicap^TM^ (the new generation of uncoated HVGICs) with different surface coating procedures and at different time points.

## 2. Materials and Methods

The G*Power 3.1.9.4 software (G*Power software, Düsseldorf, Germany) was used to calculate the total specimens. The effect size, the significance level (α), and the test power (1-β) were set at 0.4, 0.05, and 0.95, respectively. A total of 135 cylindrical specimens (6 mm in diameter and 2 mm in thickness) were prepared using metallic molds with central holes. Three types of GICs (*n* = 45/group) were tested as follows: Ketac^TM^ Universal Aplicap^TM^ (U), Ketac^TM^ Molar Aplicap^TM^ (M), and Ketac^TM^ Fil Plus Aplicap^TM^ (F). Each specimen was randomly distributed to have different surface coating procedures (*n* = 15/subgroup) as follows: uncoated (U-), coated with Ketac^TM^ Glaze (K-), and coated with G-Coat Plus (G-). The types and compositions of the GICs and the coating agents used in this study are presented in Table 1. The experimental design of this study is summarized in Figure 1.

The metallic mold was placed on a glass slide covered with a celluloid strip and filled with the test materials, following the manufacturer’s instructions. All of the materials were immediately covered with a new celluloid strip, followed by the glass slide, and pressed with a 200-g stainless steel standard weight for seven minutes to enable the early setting reaction. In the coated subgroups, the surfaces of each specimen (top, bottom, and sides) had one layer of G-Coat Plus (GF, GM, and GU subgroups) or Ketac^TM^ Glaze (KF, KM, and KU subgroups), which were applied using a disposable brush, according to the materials’ instructions. The celluloid strip was gently pressed, after which the specimen was light cured for 20 s per side using an LED curing light. The exclusion criteria were as follows: specimens with a void, a rough surface, or a surface coating thickness of more than 0.05 mm. All of the prepared specimens were submerged in individually sealed plastic vials containing 5 mL of deionized water and put in a 37 °C incubator.

The potentiometric method, which follows ISO19448-2018 [33], was performed to measure the concentrations of the released and recharged fluoride ions by using a fluoride ion-specific electrode (Orion™ 9157BNMD Triode™ 3-in-1 pH/ATC probe, Thermo Fisher Scientific Inc., Waltham, MA, USA) connected to a digital pH/ISE meter (Orion™ Versa Star Pro™, Thermo Fisher Scientific Inc., Waltham, MA, USA). The electrode was calibrated with standard fluoride solutions of 0.1, 1, 10, and 100 parts per million (ppm) mixed with a total ionic strength adjustment buffer (TISAB II with CDTA, Orion 940909, Thermo Fisher Scientific Inc., Waltham, MA, USA). Each specimen solution was prepared by pipetting 1 mL of deionized water from each specimen vial, which was then buffered with 1 mL of TISAB II. The fluoride concentration (ppm) was measured at 24 h and at weeks 1, 2, 3, and 4 of immersion.

After measuring the fluoride release for 4 weeks, all of the specimens were used to study the fluoride recharge ability. In this study, 1.23% APF gels were used for recharging the fluoride ions. Each specimen was immersed in a plastic vial containing 5 mL of the 1.23% acidulated phosphate fluoride (APF) gel for 4 min, after which the excess gel was removed by wiping it with gauze. All of the discs were left undisturbed for 30 min, after which they were washed with 50 mL of deionized water and put on an absorbent paper for 2 min to dry them. The measurement of the recharged fluoride ions was performed at the same time intervals as the fluoride release days, which were at 24 h, 1, 2, 3, and 4 weeks, respectively. After measuring the recharged fluoride at each time point, the specimens were rinsed with 50 mL of deionized water, dried on absorbent paper for 2 min, and transferred to a new, individually sealed plastic vial containing 5 mL of fresh deionized water and stored in a 37 °C incubator.

### Statistical Analysis

All of the data were analyzed using SPSS Statistics Version 22 for Windows (IBM corp., Armonk, NY, USA). Normal distribution and variance equality tests were performed using the Kolmogorov–Smirnov and Levene’s tests, respectively. The data were normally distributed but nonhomogeneous. To determine the effects of the material types, surface coating procedures, and their interactions, a two-way ANOVA followed by a post-hoc Tamhane multiple post-hoc was performed. A one-way repeated measures ANOVA and Tamhane multiple comparison test were conducted to evaluate the effects of the immersion time points. The significance level was set at 0.05.

## 3. Results

The two-way ANOVA showed that the material types, surface coating procedures, and their interactions had a statistically significant effect on the amount of released and recharged fluoride at all time points (*p* < 0.001). The mean and standard deviation of the fluoride release and recharge of the tested GICs in all subgroups and at all time points are shown in Table 2. Among the tested GICs, the Ketac^TM^ Universal Aplicap^TM^ presented the highest fluoride release at all time points, followed by the Ketac^TM^ Fil Plus Aplicap^TM^ and Ketac^TM^ Molar Aplicap^TM^, respectively, at all-time points (Figure 2a). The amount of fluoride recharge is shown in Figure 2b. Ketac^TM^ Universal Aplicap^TM^ had a statistically significant and higher level of fluoride recharge than the other two tested materials at almost all time points, except for the ones at 24 h and 4 weeks.

The effects of the surface coating procedures are shown in Figure 3. Both the Ketac^TM^ Glaze- and the G-Coat Plus-coated groups presented significantly lower fluoride release and recharge than the uncoated group at all time points (*p* < 0.001). The coating with G-Coat Plus significantly decreased the fluoride release and recharge, more than the coating with Ketac^TM^ Glaze at almost all time points (*p* < 0.05), except for the ones at 1 and 2 weeks of the fluoride release, which showed no significant differences in the fluoride ions between the G-Coat Plus- and Ketac^TM^ Glaze-coated specimens.

In Figure 4, the fluoride release and recharge patterns of each material are shown. There was a statistically significant effect on the time interval of immersion in all of the groups (*p* < 0.001). The highest amount of released fluoride occurred during the first week, decreased in the second week, and retained lower and nearly constant values over time in all of the groups. For the fluoride recharge, the highest fluoride amount occurred during the first twenty-four hours, after which it rapidly decreased in the first week and remained lower with nearly constant values over time in almost all of the test materials. Meanwhile, the Ketac^TM^ Universal Aplicap^TM^ maintained the highest fluoride recharge for one week. The peak of the fluoride recharge was lower than the peak of the fluoride release in the Ketac^TM^ Universal Aplicap^TM^ and the Ketac^TM^ Fil Plus Aplicap^TM^.

## 4. Discussion

The amount of released and recharged fluoride ions varied among the materials [34] due to differences in the resin matrix, filler composition, solubility, and porosity of the materials [23,35,36,37]. Among all of the tested GICs, Ketac^TM^ Universal Aplicap^TM^ presented the highest fluoride release and recharge at all time points. This result agreed with a previous study, which showed that materials with a high fluoride release ability also have a high fluoride recharge ability [38].

In this study, all of the specimens were prepared and tested under the same conditions by one person to eliminate human variables. However, the thickness of the coating materials, which may be one of the confounding factors, was controlled. According to the manufacturer’s fact sheet, the thickness of the G-Coat Plus coating layer is 35–40 μm. Thus, one of the exclusion criteria in this study was a coating thickness of more than 50 μm. However, no samples were excluded.

For the surface coating factor, Ketac^TM^ Glaze and G-Coat Plus significantly decreased the fluoride release and recharge in all of the test materials (Ketac^TM^ Universal Aplicap^TM^, Ketac^TM^ Molar Aplicap^TM^, and Ketac^TM^ Fil Plus Aplicap^TM^) at all time points (at 24 h, 1, 2, 3, and 4 weeks). The previous studies reported that the application of a surface coating diminished the fluoride release [23,24,25,39] and recharge [24] in some HVGICs. Kelić et al. investigated the effect of a resin coating (GC Fuji COAT LC) on the fluoride release of HVGIC (Fuji IX EXTRA) at different time points: 1 h, 24 h, 2 days, 7 days, 28 days, 84 days, and 168 days. They found that the coated specimens released fluoride at a rate about thirty times lower than the uncoated specimens [23]. Additionally, Shatat reported that resin-based coatings (G-Coat Plus and Scotchbond Universal) reduced the fluoride release ability of Ketac^TM^ Molar Aplicap^TM^, particularly during the first week [39]. Habib et al. found a dramatic reduction in the fluoride release and recharge in EQUIA Forte Fil coated with a nanofilled resin coating (EQUIA Forte Coat) [24]. This can be explained by the mechanisms of fluoride release and recharge in GICs, which consist of two parts. The first mechanism is a short-term reaction where a rapid dissolution occurs from the outer surface layer of the GICs [37,40]. The GICs dissolve several ions, such as calcium, aluminum, and fluoride ions, as the setting reaction is initiated [14]. The second mechanism is a long-term reaction where there is a slow process of fluoride ions diffusing through the bulk of the GICs into the surrounding aqueous media [37,40]. When the GIC is protected with a coating agent, the superficial layer of the immature GIC is less dissolved and, consequently, releases a smaller amount of fluoride [25]. Moreover, Shatat found that coating the GICs (Ketac^TM^ Molar Aplicap^TM^) with a nanofilled coating (G-Coat Plus) resulted in less fluoride release than coating with an unfilled resin (Riva coat) [39]. This was probably due to the micro-mechanical interlocking between the nanofiller in the G-Coat Plus and the GICs [8,11]. Moreover, another study found that an unfilled resin coating agent (Single Bond Universal Adhesive) had no micro-mechanical interlocking with the Ketac^TM^ Molar Aplicap^TM^ and the Ketac^TM^ Universal Aplicap^TM^ in SEM images [31]. In addition, hydrolysis of the unfilled resin coating occurred consistently over time and resulted in the degradation of the unfilled resin coating component over time [41].

According to the pattern of fluoride release, the highest amount of released fluoride ions occurred in the first 24 h during the initial setting reaction, called the “initial burst” [37,40]. The initial burst is important for remineralization and reducing the viability of microorganisms that may have been left in the carious dentine. After the initial burst, the fluoride release significantly decreased over the first week [42] and maintained a constant low level for 10–20 days [37]. It was reported that a small amount of fluoride is released long-term, which can last for several months to 5 years [37,40]. However, these low levels of released fluoride might be insufficient to prevent secondary caries. A fluoride concentration of at least 1 ppm is required to inhibit enamel demineralization [43]. The fluoride recharge process has the ability to take up fluoride from external sources. The capacity for fluoride recharge from external sources is very necessary to maintain the levels of fluoride [24]. The GICs act as fluoride reservoirs to increase the fluoride levels and prevent secondary caries [37].

The pattern of fluoride recharge by topical fluoride was found to be similar to the intrinsic fluoride release pattern, which consisted of two parts. The first part was a short-term increase in the first 24 h, but usually not more than the initial burst [37], and the second part was a rapid decrease to nearly pre-exposure levels within the first week [44]. Karabulut et al. evaluated the recharged fluoride ions by repeating the fluoride treatments. They found that each repetition provided a rise in the recharged fluoride ions and that they decreased in a short time [45]. Previous publications found higher fluoride releases in acidic conditions [36,46]. Ghajari et al. reported that the amount of recharged fluoride was significantly higher in 1.23% APF gels when compared to 2% NaF gels [47]. From this study, the patterns of fluoride release and recharge were slightly varied among the tested GICs; the Ketac^TM^ Universal Aplicap^TM^ presented the highest fluoride release and recharge. Moreover, we found that Ketac^TM^ Universal Aplicap^TM^ maintained the highest fluoride recharge for up to one week longer than the other tested GICs.

The limitations of the present study were that this was a short-duration in vitro study. The study did not completely reflect a clinical condition, which is influenced by various factors that are different from a laboratory situation. In oral conditions, factors, including masticatory force, brushing force, and changes in temperature and pH, can all interfere with the adhesion between the coating agents and GICs, which might affect the fluoride release and recharge abilities of the GICs. Further clinical studies or laboratory-designed, simulated oral conditions should be conducted. Moreover, a further long-term study (3–6 months) should be performed to evaluate the amount of fluoride. In addition, home-use fluorides, such as fluoride dentifrice and fluoride mouth rinse, which are important sources of external fluoride, should be investigated.

The clinical decisions between coated and uncoated materials depend on the purpose for which these materials are chosen. To maximize the remineralization effects, uncoated materials might be a better choice. However, maintaining oral hygiene is still an important factor to enhance remineralization and optimize fluoride reservoirs for the GICs.

## 5. Conclusions

Within the limitations of this study, the application of coating agents, especially nanofilled resin coatings, reduced the fluoride release and recharge of Ketac^TM^ Universal Aplicap^TM^.

## Figures and Tables

**Figure 1 dentistry-10-00233-f001:**
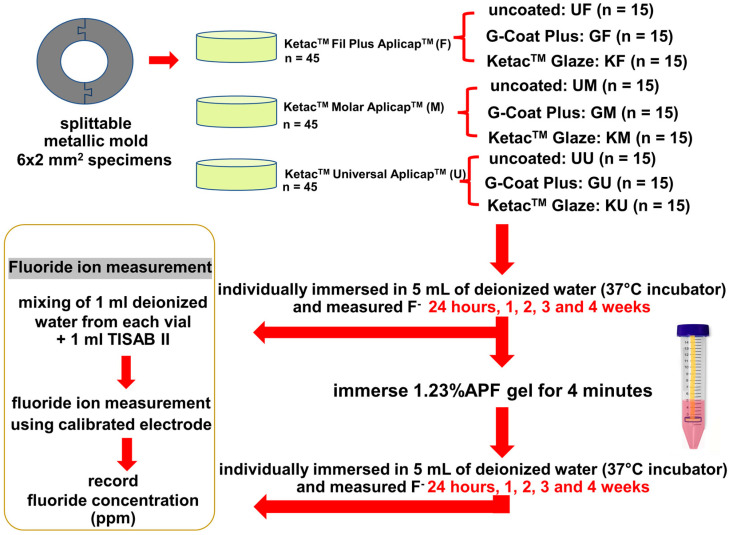
The experimental design. F = Ketac^TM^ Fil Plus Aplicap^TM^ uncoated (UF), coated with G-Coat Plus (GF), and coated with Ketac^TM^ Glaze (KF); M = Ketac^TM^ Molar Aplicap^TM^ uncoated (UM), coated with G-Coat Plus (GM), and coated with Ketac^TM^ Glaze (KM); U = Ketac^TM^ Universal Aplicap^TM^ (U) uncoated (UU), coated with G-Coat Plus (GU), and coated with Ketac^TM^ Glaze (KU).

**Figure 2 dentistry-10-00233-f002:**
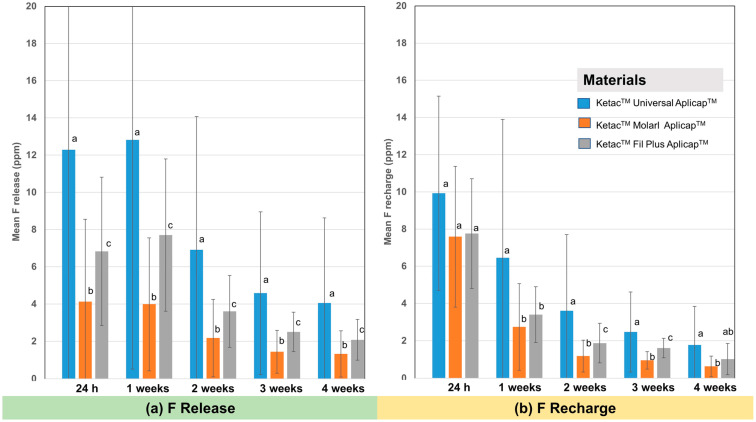
Means of the amount of fluoride (ppm) in each material (Ketac^TM^ Universal Applicap^TM^, Ketac^TM^ Molar Applicap^TM^, and Ketac^TM^ Fill Plus Applicap^TM^) at 24 h and at 1, 2, 3, and 4 weeks of (**a**) fluoride release and (**b**) fluoride recharge. The different letters on each bar represent statistically significant differences (*p* < 0.05) within each time point.

**Figure 3 dentistry-10-00233-f003:**
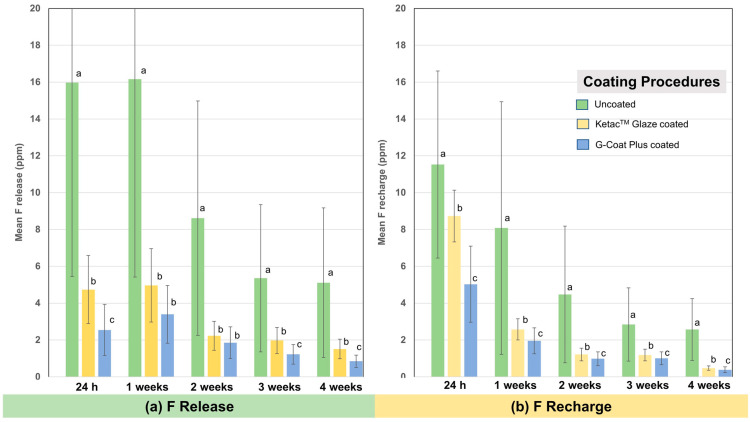
Means of the amount of fluoride (ppm) for each surface coating procedure (uncoated, Ketac^TM^ Glaze-coated, and G-Coat Plus-coated) at 24 h and at 1, 2, 3, and 4 weeks of (**a**) fluoride release and (**b**) fluoride recharge. The different letters on each bar represent statistically significant differences (*p* < 0.05) within each time point.

**Figure 4 dentistry-10-00233-f004:**
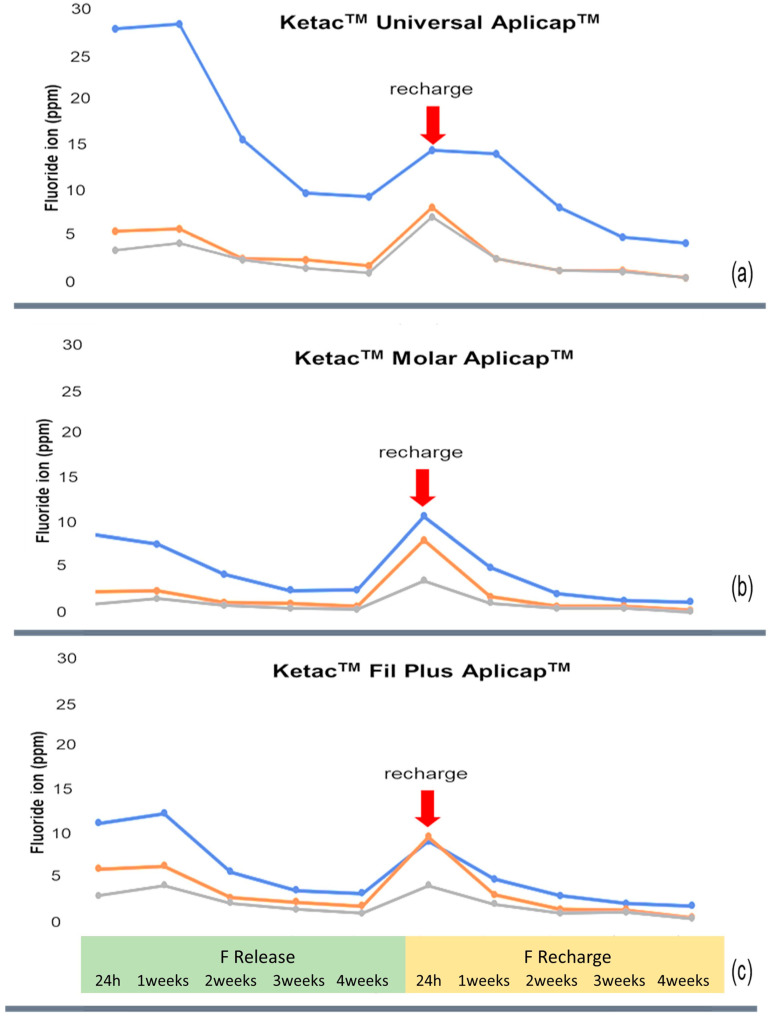
Means of the amount of fluoride (ppm) for each surface coating procedure and each type of material: (**a**) Ketac^TM^ Universal Aplicap^TM^, (**b**) Ketac^TM^ Molar Aplicap^TM^, and (**c**) Ketac^TM^ Fil Plus Aplicap^TM^ at 24 h and at 1, 2, 3, and 4 weeks of fluoride release and recharge. The recharge point was before the fluoride release measurement, and the fluoride recharge measurement was after exposure to a 1.23% APF gel.

**Table 1 dentistry-10-00233-t001:** Composition of the materials that were used in this study.

Material	Composition
Ketac^TM^ Fil Plus Aplicap^TM^ (CGICs)	Powder: glass powder (>99 wt%) Liquid: water (40–55 wt%), copolymer of acrylic acid–maleic acid (35–55 wt%), tartaric acid (5–10 wt%) Powder/liquid ratio: 3.0/1.0 wt
Ketac^TM^ Molar Aplicap^TM^	Powder: glass powder (93–98 wt%) Liquid: water (60–65 wt%), copolymer of acrylic acid–maleic acid (30–40 wt%), tartaric acid (5–10 wt%) Powder/liquid ratio: 3.4/1.0 wt
Ketac^TM^ Universal Aplicap^TM^ (HVGICs)	Powder: oxide glass (>95 wt%) Liquid: water (40–60 wt%), copolymer of acrylic acid–maleic acid (30–50 wt%), tartaric acid (1–10 wt%), benzoic acid (<0.2 wt%) Powder/liquid ratio: 3.2/1.0 wt
Ketac^TM^ Glaze (Unfilled resin coating agent)	Dicyclopentyldimethylene diacrylate (>95 wt%), [(3-methoxypropyl)imino]di-2,1-ethanediyl bismethacrylate (1–5 wt%), 2-[(2-hydroxyethyl)(3-methoxypropyl)amino]ethyl methacrylate (<1 wt%), 2,2- dimethoxy-1,2-diphenylethan-1-one (<0.5 wt%)

**Table 2 dentistry-10-00233-t002:** Means ± standard deviation (SD) of the fluoride release and recharge of the tested GICs.

Materials	Coating	Fluoride Release					Fluoride Recharge				
		24 h	1 Week	2 Weeks	3 Weeks	4 Weeks	24 h	1 Week	2 Weeks	3 Weeks	4 Weeks
Ketec^TM^ Universal Aplicap^TM^	Uncoated	27.73 ± 9.35 ^Aa^	28.27 ± 9.35 ^Aa^	15.69 ± 6.05 ^Ab^	9.79 ± 3.92 ^Ac^	9.35 ± 4.47 ^Ac^	14.43 ± 6.89 ^ABb^	14.13 ± 8.88 ^Abc^	8.18 ± 4.36 ^Ad^	4.91 ± 2.17 ^Ae^	4.33 ± 1.72 ^Af^
	Ketac^TM^ Glaze	5.63 ± 1.17 ^Ba^	5.92 ± 1.48 ^BFa^	2.58 ± 0.57 ^BFb^	2.47 ± 0.6 ^Bb^	1.8 ± 0.36 ^Bc^	8.19 ± 0.77 ^ACd^	2.67 ± 0.29 ^Bb^	1.37 ± 0.23 ^BCe^	1.29 ± 0.2 ^BCf^	0.51 ± 0.1 ^BCg^
	G-Coat Plus	3.49 ± 1.12 ^Ca^	4.29 ± 1.32 ^BGa^	2.47 ± 0.75 ^BFb^	1.52 ± 0.45 ^CDEc^	1.02 ± 0.3 ^Cd^	7.18 ± 2.13 ^Ce^	2.57 ± 0.61 ^BGHb^	1.28 ± 0.35 ^BDf^	1.22 ± 0.33 ^BCf^	0.5 ± 0.15 ^BCg^
Ketec^TM^ Molar Aplicap^TM^	Uncoated	8.95 ± 5.13 ^BEa^	7.79 ± 4.01 ^BFEGa^	4.39 ± 2.34 ^BEFb^	2.61 ± 1.35 ^BEFGc^	2.66 ± 1.37 ^BDc^	10.88 ± 3.66 ^ABCd^	5.15 ± 2.71 ^ABEFb^	2.18 ± 0.79 ^CEc^	1.43 ± 0.51 ^BCe^	1.31 ± 0.45 ^Df^
	Ketac^TM^ Glaze	2.48 ± 0.64 ^Ca^	2.53 ± 0.35 ^Ca^	1.25 ± 0.15 ^Cb^	1.12 ± 0.12 ^Cc^	0.83 ± 0.95 ^Cd^	8.2 ± 1.67 ^ABCe^	1.91 ± 0.21 ^Cf^	0.78 ± 0.06 ^Fd^	0.8 ± 0.09 ^Dd^	0.35 ± 0.03 ^Eg^
	G-Coat Plus	1.08 ± 0.38 ^Dae^	1.65 ± 0.36 ^Db^	0.89 ± 0.19 ^Da^	0.61 ± 0.13 ^Hc^	0.5 ± 0.16 ^Ec^	3.69 ± 0.71 ^Dd^	1.19 ± 0.17 ^De^	0.59 ± 0.1 ^Gc^	0.61 ± 0.09 ^Ec^	0.21 ± 0.03 ^Ff^
Ketec^TM^ Fill Plus Aplicap^TM^	Uncoated	11.33 ± 3.26 ^Ea^	12.43 ± 3.56 ^Eb^	5.76 ± 1.94 ^Ec^	3.67 ± 0.96 ^Fd^	3.33 ± 0.9 ^Dd^	9.27 ± 2.3 ^ABCe^	4.95 ± 1.56 ^Ef^	3.05 ± 1.08 ^Ed^	2.19 ± 0.52 ^Fg^	1.91 ± 0.88 ^Dg^
	Ketac^TM^ Glaze	6.09 ± 0.83 ^Ba^	6.45 ± 0.67 ^Fa^	2.85 ± 0.22 ^Bb^	2.33 ± 0.15 ^Bc^	1.9 ± 0.13 ^Bd^	9.79 ± 0.99 ^Be^	3.15 ± 0.27 ^FGf^	1.49 ± 0.12 ^Bcg^	1.46 ± 0.12 ^Bg^	0.57 ± 0.06 ^Bh^
	G-Coat Plus	3.06 ± 1.06 ^Ca^	4.24 ± 0.98 ^BGb^	2.21 ± 0.44 ^Fa^	1.53 ± 0.27 ^DGc^	1.02 ± 0.19 ^Cd^	4.21 ± 0.88 ^Db^	2.11 ± 0.32 ^CHa^	1.08 ± 0.17 ^Dd^	1.18 ± 0.14 ^Cd^	0.43 ± 0.07 ^Ce^

The different uppercase and lowercase letters represent statistically significant differences for each column and row, respectively (*p* < 0.05).

## Data Availability

Not applicable.

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
