# Peer review of "Effects of Protective Surface Coating on Fluoride Release and Recharge of Recent Uncoated High-Viscosity Glass Ionomer Cement"

_dentistry, 2022, doi:10.3390/dj10120233_

Round 1

Reviewer 1 Report

First of all, I would like to congratulate the authors on their choice of topic. As the ability to release and recharge fluoride is one of the most interesting properties of glass ionomer cements, this is a very interesting and important topic, especially in paediatric dentistry.

​What should be changed, however, are Figures 3 and 4. t is completely unclear to the reader what the above letters stand for, the result of ANOVA or a post hoc test. In my opinion, it is not necessary to indicate the exact amount of ppm. the value on the y-axis is sufficient.  If the authors found it  complicated, please present the results in the form of a Table.

Reviewer 2 Report

Dear authors,

thank you for your study. Please correct your manuscript in accordance to the following comments.

Page 2, lines 60-70: Information provided is not entirely clear. If Ketac Universal Aplicap does not need a coating, why the influence of different coatings should be observed for this material? The paragraph is confusing. Please correct this issue. Information provided might cause misunderstanding.

Please increase contrast and sharpness of figure 1. Also erase the cut-offs in the top left corner!

Please provide more detailed information upon the process of fluoride recharging. What was the purpose? Why this measurement was included? This is a very important issue. Please clearly address this issue in your manuscript.

Establish a single section "Statistical analysis".  Include the analysis from page 2 lines 72-74.

Figure 2: Please increase resolution of your images. Consider to apply box-plots. Please include standard divisions (SD). Please describe the letters "a-c" in the figures! Provide headlines!

Figure 3: Also increase resolution. See remarks given for figure 2 too. The difference between figure 2 and 3 is not obvious. Please address this issue! Please provide headlines for each diagram.

Figure 4: Increase resolution. Correct length and width of the figure. What is the meaning of "recharge"? What happens at this point? Please provide detailed information upon the recharging process in the introduction and M&M section too.

Round 2

Reviewer 2 Report

Dear authors,

thank you for correcting your manuscript as adviced. There are no further revisions necessary.